# The Effects of Fisetin on Cyclosporine-Treated Dry Eye Disease in Dogs

**DOI:** 10.3390/ijms24021488

**Published:** 2023-01-12

**Authors:** Kristína Krajčíková, Agnieszka Balicka, Mária Lapšanská, Alexandra Trbolová, Zuzana Guľašová, Daria Kondrakhova, Vladimír Komanický, Adriana Rašiová, Vladimíra Tomečková

**Affiliations:** 1Department of Medical and Clinical Biochemistry, Faculty of Medicine, Pavol Jozef Šafárik University in Košice, 040 11 Košice, Slovakia; 2Small Animal Clinic, University Veterinary Hospital, University of Veterinary Medicine and Pharmacy in Košice, Komenského 73, 041 81 Košice, Slovakia; 3Department of Experimental Medicine, Faculty of Medicine, Pavol Jozef Šafárik University in Košice, Trieda SNP 1, 040 11 Košice, Slovakia; 4Institute of Physics, Department of Condensed Matter Physics, Faculty of Science, Pavol Jozef Šafárik University in Košice, Park Angelinum 9, 041 54 Košice, Slovakia

**Keywords:** dry eye syndrome, fisetin, MMP-9, polyphenols, tear film production

## Abstract

Dry eye disease (DED) is a chronic debilitating ophthalmological disease with the current therapeutic options focused on the suppression of the symptoms. Among the possibilities of how to improve DED therapy, polyphenols have shown an enormous capacity to counteract DED functional changes. The study aimed to specifically target pathophysiological mechanisms by the addition of fisetin to the cyclosporine treatment protocol. We examined dog patients with DED on cyclosporine treatment that were administered 0.1% fisetin or fisetin-free eye drops. For the assessment of fisetin effects, tear film production and matrix metalloproteinase 9 (MMP-9) were studied in the tear film. Tear production was not recovered after 7 or 14 days (9.40 mm ± 6.02 mm, *p* = 0.47; 9.80 mm ± 6.83 mm, *p* = 0.53, respectively). MMP-9 levels significantly increased after 7 days and then dropped after 14 days (775.44 ng/mL ± 527.52 ng/mL, *p* = 0.05; 328.49 ng/mL ± 376.29 ng/mL, *p* = 1.00, respectively). Fisetin addition to cyclosporine DED treatment was not able to restore tear fluid production but influenced molecular pathological events through MMP-9.

## 1. Introduction

Dry eye disease (DED, dry eye syndrome, *keratoconjunctivitis sicca*) is defined as “a multifactorial disease of the ocular surface characterized by a loss of homeostasis of the tear film and accompanied by ocular symptoms, in which tear film instability and hyperosmolarity, ocular surface inflammation and damage, and neurosensory abnormalities play etiological roles” [1]. The risk factors include age [2], sex (especially postmenopausal and hormonal replacement therapy women) [3,4], several systemic disease medications [5], contact lens use [6], and a few other factors [7,8,9]. Furthermore, the modern work environment represents a challenge for eye health due to air conditioning and intense computer use [10].

The key sign is hyperosmolarity of the tear film, which leads to a DED-specific pathological inflammatory cascade: The activation of activator protein 1 (AP-1) and nuclear factor-κB (NF-κB) transcription factors, mitogen-activated protein kinase (MAPK) p38, c-Jun N-terminal kinase 1 and 2 (JNK1/2), and extracellular-regulated kinase 1 and 2 (ERK1/2) [1,11,12,13]. These transmit the signal to the production of inflammatory cytokines, matrix metalloproteinases (MMPs), and other chemokines triggering ocular surface damage with subsequent worsening of tear film homeostasis. Furthermore, inflammation, extracellular matrix remodeling, and persistent hyperosmolar conditions lead to the continuous production of reactive oxygen species (ROS) inducing a decrease in antioxidant enzymes, thus disrupting the fine-tuned redox homeostasis [14].

DED treatment is a stepwise procedure. In the initial stages, it is focused on symptom amelioration by applying artificial tears, which, however, do not address the causes or pathophysiological mechanism. In more severe cases, various topical drugs are used, primarily corticosteroids, antibiotics, or cyclosporine, which effectively target the inflammatory response. Although their use is effective, they could not break the chronic nature of DED and have various limitations, such as short duration, poor bioavailability, or side effects [15]. For instance, blepharospasms, corneal edema, discomfort, irritation, and decreased blood lymphocyte levels were detected in patients treated with cyclosporine [16,17]. Moreover, it needs to be administered frequently due to the chronic nature of the disease. For these reasons, it is essential to seek effective additional therapy. The common complementary active ingredients are of natural origins, such as ω-3 fatty acids [18], propolis, aloe vera, chamomile, and melon [19,20], which showed encouraging therapeutic potential. The active ingredients of these sources are mainly polyphenols, the effect of which has been the subject of many in vitro and in vivo studies in the treatment of DED [21]. Finding new therapies is inevitable for the management of DED patients. Therefore, we studied the effect of fisetin addition to cyclosporine-treated canine DED to target oxidative and inflammatory pathways at one time.

Fisetin is a 3, 7, 3′, 4′—tetrahydroxyflavone with antioxidant [22], anti-inflammatory [23], neuroprotective [24], antiaging [25], and antiviral [26] effects. It has an impact on angiogenesis [27] and glucose metabolism [28], the most robust inhibition of lipid peroxidation in lung homogenates compared to the other seven studied polyphenols [29], and exhibited the most potent senolytic activity compared to the other 10 studied polyphenols [25]. In eye diseases, fisetin protects the retinal epithelium against oxidative stress [30] and suppresses the inflammatory response [31]. Furthermore, fisetin treatment of the mouse glaucoma model also showed amelioration of pattern electroretinogram, visually evoked potentials, intraocular pressure, retinal ganglion cell survival, and microglia activation [32]. The protective effects were mediated by the nuclear factor erythroid 2-related factor (Nrf-2) that regulates the transcription of many genes of antioxidant defense [30] together with suppression of the activity of the cAMP response element-binding protein (CREB) that is involved in redox balance and the inflammatory response, and specific kinases: MAPK p38, JNK1/2, and ERK1/2 [31]. To date, fisetin’s effects against DED have not been studied. However, other structurally similar polyphenols exhibited promising therapeutic effects in vitro and in vivo in numerous studies [21].

This pilot study, for the first time, investigates its complementary therapeutic potential in cyclosporine-treated DED assessing tear production and MMP-9 levels.

## 2. Results

### 2.1. Study Design and Population Characteristics

In the study, a total of 17 participants (26 eyes) were enrolled and examined at the Small Animal Clinic at the University of Veterinary Medicine and Pharmacy in Košice, from which there were eight participants (16 eyes) that met the criteria for the healthy group and nine participants (10 eyes) for the dry eye disease group. Participants in the dry eye disease group were randomly assigned to intake 0.1% fisetin in hypromellose eye drops (five participants, five eyes) or control-hypromellose eye drops without fisetin (four participants, five eyes) for 14 days. One eye from the healthy control group was excluded from the study due to unacceptable laboratory values (Figure 1). The distribution of age, gender, and disease characteristics are summarized in Table 1. The age of the fisetin group was significantly higher when compared to the healthy group due to a higher incidence of DED among the older subjects on account of a decline in tear film production with higher age [33]. Moreover, the mean age of the control group was higher in comparison with healthy subjects, but it did not reach statistical significance. Although there was an uneven distribution of sex, this parameter does not influence tear production [33]. The participants were of various breeds.

### 2.2. Atomic Force Microscopy

In addition to the standard diagnostic methods for DED, we used AFM as an experimental diagnostic tool. AFM enables the observation of different tear film compositions by scanning and displaying the crystals of tear film components. Indeed, the healthy group showed a continuous network of heart-shaped crystals with an average crystal size of 0.5–1 µm. Large crystals had well-pronounced longitudinal symmetry. The surface roughness was 38.2 nm over the entire area of the sample. The average height of the dendrites was 75–85 nm, and the length of the dendritic branches was more than 90 μm (Figure 2A). In contrast to this, the dry eye group showed a fern-like structure, highly branched along the length, consisting of small crystals. The width of the branches increased over the entire study area from the beginning to the end. The surface roughness did not exceed 70 nm. The main branches of dendrites were 80 µm long. The width of small branches on the main branches averaged 1–2 µm, and the length was 20 µm. The average height of the dendrites was approximately 200 nm (Figure 2B).

### 2.3. Tear Film Production

The Fisetin and control groups significantly differed in tear film production in comparison with healthy (8.00 mm ± 5.10 mm, *p* < 0.0001; 10.60 mm ± 3.21 mm, *p* < 0.0001, respectively, in comparison with 20.07 mm ± 2.94 mm) at the beginning of the experiment, indicating eligible subjects in the dry eye group. No significant amelioration in tear production was observed in the fisetin group after 7 or 14 days (9.40 mm ± 6.02 mm, *p* = 0.47; 9.80 mm ± 6.83 mm, *p* = 0.53, respectively). Further analysis showed no statistically significant effect of fisetin when compared with the vehicle (9.40 mm ± 6.02 mm vs. 11.00 mm ± 2.55 mm, *p* = 0.60 at day 7, 9.80 mm ± 6.83 mm vs. 9.40 mm ± 4.72 mm, *p* = 0.92 at day 14) (Figure 3).

### 2.4. Tear Film MMP-9 Level

Fisetin and control groups did not significantly differ in MMP-9 tear film levels at the beginning of the experiment (fisetin: 354.83 ng/mL ± 348.31 ng/mL, control: 247.261 ng/mL ± 48.91 ng/mL). However, MMP-9 increased significantly in the fisetin group after 7 days (775.44 ng/mL ± 527.52 ng/mL, *p* = 0.047) and subsequently decreased after 14 days below the values at the beginning of the experiment (328.49 ng/mL ± 376.29 ng/mL) (Figure 4). The control group showed very few insignificant changes over time (219.60 ng/mL ± 32.09 ng/mL after 7 days, 229.08 ng/mL ± 48.03 ng/mL after 14 days).

### 2.5. The Dependence of MMP-9 on Tear Production

The tear MMP-9 levels showed a negative moderate correlation with tear production in the fisetin group (r = −0.44, *p* = 0.10) (Figure 5). On the other hand, the control and healthy groups showed a positive correlation (r = 0.41, *p* = 0.13; r = 0.13, *p* = 0.64, respectively). The opposite trend of the fisetin group, together with the weak correlation of the healthy group, highlights the impact of fisetin on MMP-9 levels and improvements in tear film production.

## 3. Discussion

This experimental study investigated the effect of fisetin addition to DED treatment with cyclosporine and artificial tears (hypromellose eye drops) on the restoration of tear production and MMP-9 levels in dogs.

The MMP-9 biomarker is valid in indicating ocular surface inflammation and DED severity, and it is a relevant therapeutic target [34]. Its level decreased after topical application of polyphenol quercetin [35] or pterostilbene [36] in vivo and in vitro in DED studies. The main finding was that fisetin caused an increase in MMP-9 after 7 days with a subsequent decrease after 14 days. We assume the immediate increase was the result of a damaged cornea and the conjunctiva healing process, which is accompanied by inflammation, hence the MMP-9 increase. The following MMP-9 decrease below the starting point value indicates fisetin’s potential to further decrease MMP-9 in DED patients. This is in accordance with the encouraging results from in vivo studies in which structurally very similar compounds—from the same polyphenol group of flavones—were used. Specifically, 0.1% resveratrol administered 3 times per day decreased pro-inflammatory cytokines and reversed CD4+ T cell infiltration from day 6 [37]. Furthermore, 0.01% quercetin administered 3 times per day improved the corneal surface [37].

In both cases, they did not restore tear film production, which is also consistent with the findings of our study. However, 0.5% quercetin administered 4 times per day significantly restored the tear film after 3 days together with decreasing MMP-9, MMP-2, tumor necrosis factor α (TNF-α), intercellular adhesion molecule 1 (ICAM-1), and vascular cell adhesion molecule 1 (VCAM-1) [35]. Moreover, in a double-blind randomized controlled clinical trial with green tea extract (rich in polyphenols, mainly epigallocatechin gallate) topically applied for one month, no significant difference was observed in the restoration of tear film production and corneal and conjunctival staining [38]. Additionally, the patients were without any ophthalmic medication for at least one month before the study. Based on these results, it seems that polyphenols might not be very efficient in the restoration of tear film production, but rather in tear film stability [38].

The environmental and other unknown factors influencing tear film homeostasis and interindividual reactions to them are also likely to be the reason for a very weak and insignificant correlation between MMP-9 level and tear film production. Considering this, the MMP-9 response to environmental stimuli should be further evaluated to capture the potential confounding factors for future studies.

One of the limitations of the study was its pilot character manifested in the sample size. Secondly, treatment with cyclosporine before the beginning of the study might have affected the ability to capture the direct effect of fisetin on the restoration of tear film production and MMP-9 level. Thirdly, the subjects did not cooperate in tear collection equally, which may have affected eye irritation and MMP-9 production. Despite this, fisetin showed minimal potential in tear film production but reduced MMP-9.

To summarize, finding an effective treatment for DED is a necessity in terms of patients’ quality of life. A growing number of studies revealed the huge potential of polyphenols. This study demonstrates the potential of polyphenol fisetin addition to cyclosporine-treated DED. Tear film production was not improved during the 2 weeks of the fisetin treatment, but the subjects benefited from the fisetin application by decreasing tear film MMP-9.

## 4. Materials and Methods

### 4.1. Study Design

The randomized, open-label animal trial between fisetin eye drops and the control was performed to evaluate its effect on tear film production and the tear film MMP-9 biomarker in dogs treated with cyclosporine. The healthy control group was included in the study and served as an internal comparator.

The study was performed at the Small Animal Clinic, University of Veterinary Medicine and Pharmacy in Košice, Slovakia. The participants were privately owned pet dogs, patients of the Small Animal Clinic of the University of Veterinary Medicine and Pharmacy in Košice. Dogs were vaccinated and dewormed following the schedules. Food was given to the dogs in different qualities and quantities following owners’ preferences. Water was given ad libitum. The body condition score (BCS) was 4–5/9 in all cases. The research was reviewed and approved by the Scientific Research Committee of the University of Veterinary Medicine and Pharmacy in Košice (No. EKY8/2021-3) concerning non-experimental clinical patients. The owners were informed about the details of the study and gave their verbal and written consent. The study was performed in accordance with Directive 2010/63/EU of the European Parliament and of the Council of 22 September 2010 on the protection of animals used for scientific purposes, Chapter I, Article 1, point 5(b).

After informed consent was obtained, participants who met all eligibility requirements were randomly assigned to the following groups:

Fisetin group: Treatment consisted of 0.1% fisetin in Hypromelóza—P (Unimed Pharma, Slovakia) eye drops (artificial tears) 3 times per day via 1 drop bilaterally for 14 days. Hypromelóza—P was used to ensure fisetin stability [39]. The fisetin dosage and duration of treatment were chosen based on the previous in vivo studies with polyphenols from the structurally same group—flavones [21]. The dose of fisetin administered was not related to the subject’s weight as in the case of ophthalmic solution administration, tolerance of the ocular surface and frequency of application is overriding [40].

Control group: Treatment consisted of Hypromelóza—P (Unimed Pharma, Slovakia) eye drops (artificial tears) 3 times per day via 1 drop bilaterally for 14 days.

Healthy group: No treatment; The group consisted of healthy dogs based on physical examination with no history of vision impairment or ocular abnormality.

The positive control group was omitted due to the lack of clinical studies.

The tear film of participants was obtained on days 0, 7, and 14.

### 4.2. Participants

Nine different breed dogs (weight: 8–30 kg) with long-lasting DED (keratoconjunctivitis sicca) treated with cyclosporine eye drops were enrolled in the study. The inclusion criterium was DED diagnosis based on the value of the Schirmer test less than 10 mm/min and Tear breakup time (TBUT) less than 7 s at first examination. All patients had mild mucopurulent discharge and conjunctival hyperemia. Although dry and lusterless cornea vascularization and pigmentation were observed, in none of the cases was fluorescein staining positive. A total of eight different breed dogs (weight: 3–30 kg) were included in the study as a healthy group. All animals were healthy dogs based on physical examination with no history of vision impairment or ocular abnormality. The dogs were screened by distant and close ophthalmic examination, including neuro-ophthalmic and vision tests. Ocular examination was performed using the Schirmer test (STT, Eickemeyer, Tuttlingen, Germany), fluorescein staining—fluorescein Sodium Ophthalmic Strips (OptiTech, Prayagraj, India), slit-lamp biomicroscopy (Kowa SL15, Kowa, Tokyo, Japan), rebound tonometry (TonoVet, iCare, Vantaa, Finland), and indirect funduscopy (Keeler, Windsor, UK).

### 4.3. Tear Film Collection and Extraction

Tear samples were collected from the temporal aspect of the ventral conjunctival fornix with a Schirmer strip (STT, Eickemeyer, Germany) for tear fluid production and MMP-9 level measurements, with the flushing method using 100 µL the saline solution for AFM measurements. Schirmer strip samples were further processed and put into the microcentrifuge tubes (V = 200 µL) with 100 µL of the Halt antiprotease inhibitor cocktail (Pierce, Rockford, IL, USA), kept, transferred on ice, and stored at −80 °C until analysis. Tear film from the strip was recovered directly before analysis by centrifugation at 21,000× *g* for 3 min at 4 °C. Before centrifugation, the microcentrifuge tubes were pierced at the bottom, put into the larger tubes (V = 1.5 mL) with the plastic pipette tip, and then centrifugated. Samples attained by flushing were put into the microcentrifuge tubes, kept, transferred on ice, and applied on the microscopic glass slides.

### 4.4. MMP-9 Enzyme-Linked Immunosorbent Assay

An Enzyme-Linked Immunosorbent Assay (ELISA) was carried out to detect canine MMP-9 concentrations in the tear film of all groups using the Dog Matrix Metalloproteinase 9 (MMP9) ELISA Kit (Abbexa Ltd., Cambridge, UK) according to the manufacturer’s instructions. Absorption was measured on the Synergy™ H4 Hybrid Multi-Mode Microplate Reader (BioTek, Friedrichsthal, Germany) at a wavelength of 450 nm. Standard curves were constructed from the standard solutions included in the kit.

### 4.5. Atomic Force Microscopy

Atomic force microscopy (AFM) was used as an experimental diagnostic tool as described elsewhere [41,42]. Briefly, 2 μL of tear fluid was deposited on the microscopic glass slides and stretched over the surface by the method of blood smear. The slides were dried at room temperature without a fixative. The samples were analyzed using the atomic force microscope Dimension Icon^®^ (ICON, Bruker, Berkley, CA, USA) in tapping mode with silicon tips (MikroMasch, Berkley, CA, USA, NSC35 series) with a radius of curvature of 10 nm. The surface of each sample was processed by ScanAsyst™ software.

### 4.6. Statistical Analysis

No sample size calculations were carried out due to the pilot exploratory nature of the study and the unknown vehicle effect. To analyze the normality of data distribution, the Shapiro–Wilk test was used. The unpaired two-sided *t*-test was applied for the analysis of differences between the groups in age, tear film production, and MMP-9 level. The paired two-sided *t*-test was used for the comparison of tear film production and MMP-9 level in one group at various times. The relationship between MMP-9 and tear film production was analyzed using Pearson’s correlation.

## Figures and Tables

**Figure 1 ijms-24-01488-f001:**
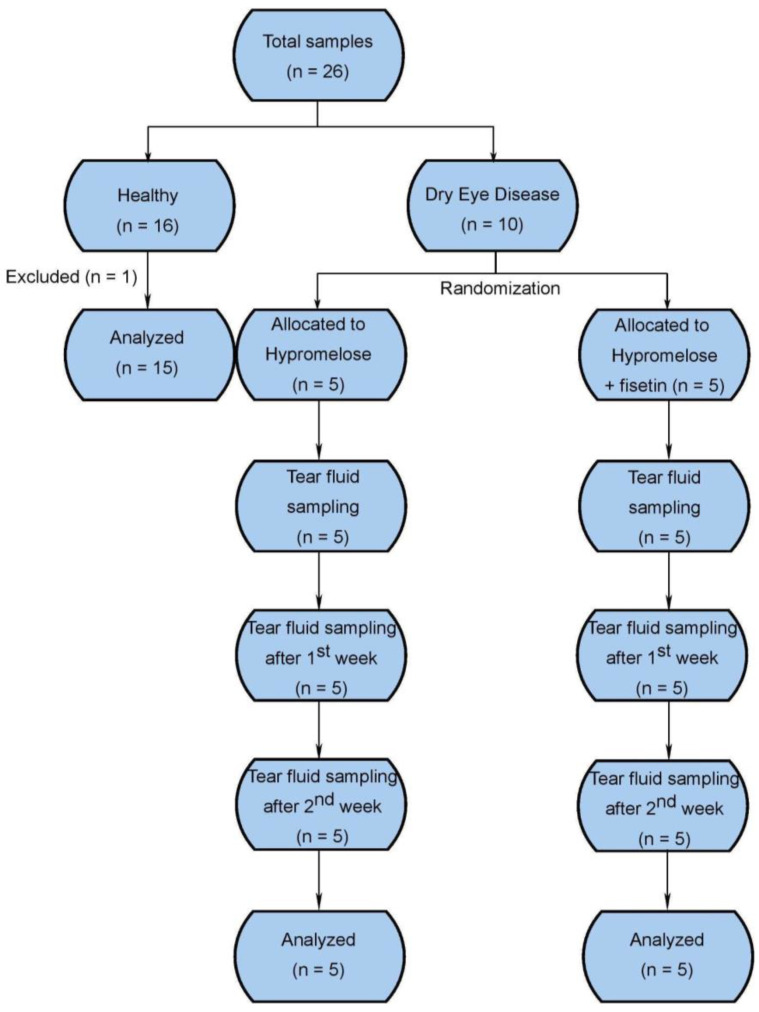
Design of the clinical study; *n*—number of eyes.

**Figure 2 ijms-24-01488-f002:**
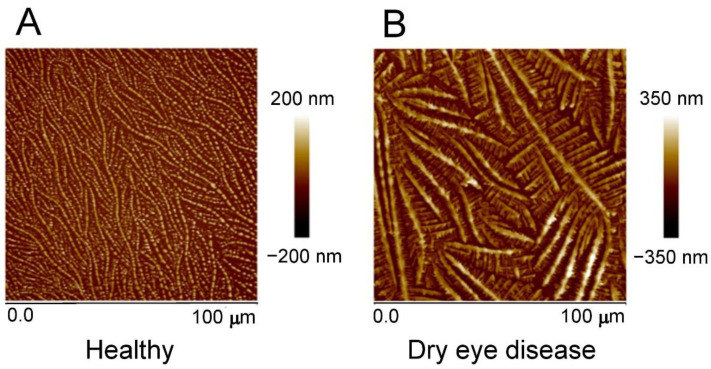
Tear film crystals after dehydration in the group of (**A**) healthy subjects and (**B**) dry eye disease subjects.

**Figure 3 ijms-24-01488-f003:**
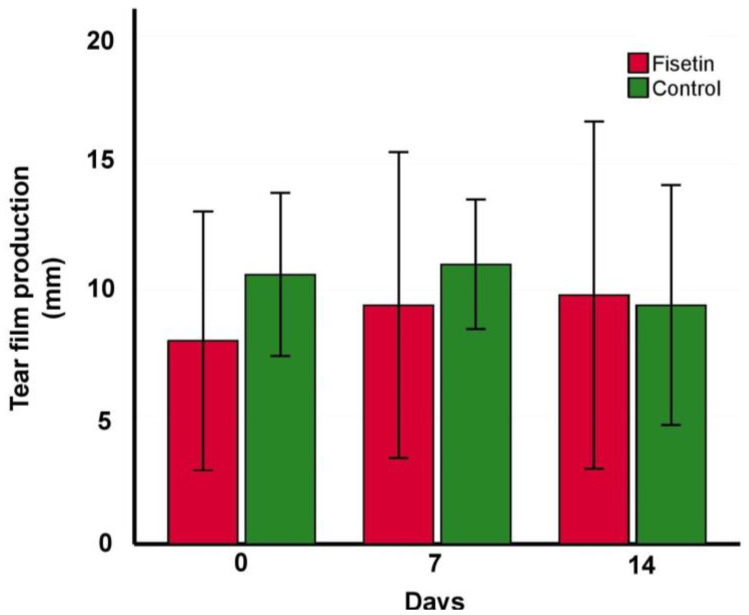
Mean change in tear film production over the treatment period in fisetin (red) and control (green) groups. Statistical significance was tested using an unpaired two-sided *t*-test.

**Figure 4 ijms-24-01488-f004:**
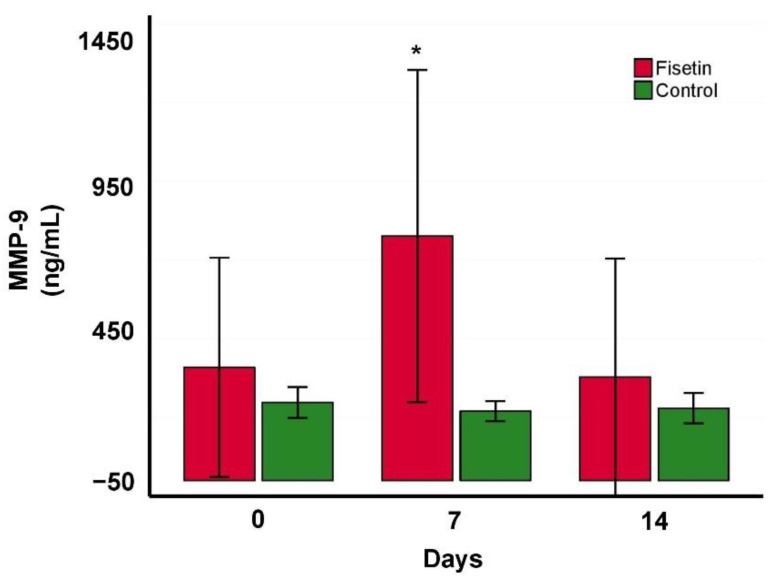
Mean change in tear film MMP-9 levels over the treatment period in fisetin (red) and control (green) groups. Statistical significance was tested using an unpaired two-sided *t*-test, * *p* <0.05.

**Figure 5 ijms-24-01488-f005:**
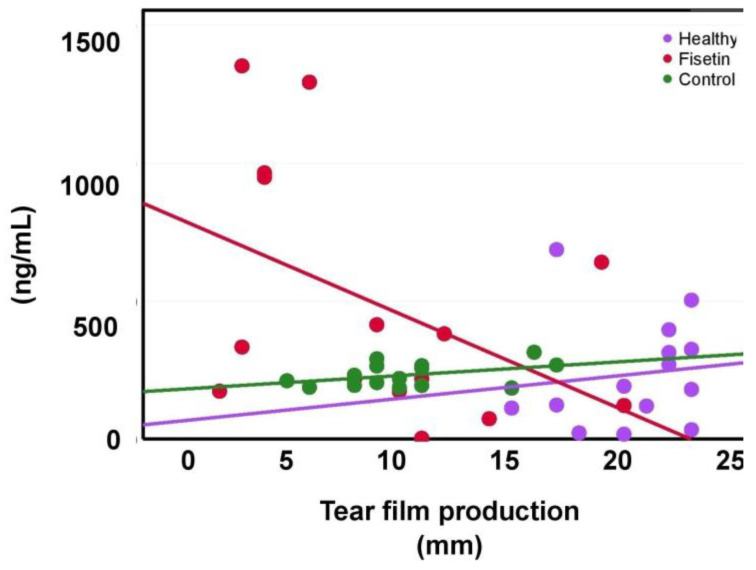
Scatter plot showing the relationship between matrix metalloproteinase 9 (MMP-9) in healthy (purple), fisetin (red), and control (green) groups and tear production. Statistical significance was tested using Pearson’s correlation.

**Table 1 ijms-24-01488-t001:** Demographics and Baseline Characteristics.

Variable	Healthy (*n* = 8)	0.1% Fisetin	Control
		(*n* = 5)	(*n* = 4)
Sex, *n* (%)			
XY	5 (62.5%)	4 (80%)	0 (0%)
XX	3 (37.5%)	1 (20%)	4 (100%)
Age			
Mean (years)	4.54 ± 3.28	11.60 ± 1.67 *	8.25 ± 3.69
Fluorescein staining	-	-	-

Values are expressed as a percentage or mean ± standard deviation. * The significant differences between fisetin and the healthy group, *p* = 0.0005, were analyzed by *t*-test.

## Data Availability

Not applicable.

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
