# Peer review of "The Effects of Fisetin on Cyclosporine-Treated Dry Eye Disease in Dogs"

_ijms, 2023, doi:10.3390/ijms24021488_

Round 1

Reviewer 1 Report

In this interventional study in dog subjects with dry eye disease, the authors tried to evaluate the efficacy of cyclosporine eye drops with fisetin addition and compare it with fisetin-free cyclosporine eye drops. The study was presented in a clear way that facilitated quick detailed assessments. There are a few critical issues I'd like to point out for further improvements. However, the study design and sample size significantly limit the authors' ability to provide solid answers to their research questions.

Major issues

1. The randomization was unsuccessful, resulting in an imbalance between the fisetin and cyclosporine-only groups. This could also be due to the sample size issue in the next point.

2. The sample size did not provide sufficient statistical power in the analysis. We can tell from the large and mostly overlapping confidence intervals in the statistics and plots.

3. The control group was not followed up in the last two time points. Please provide a scientific reason to justify this design. With the current design, the authors will not be able to evaluate the efficacy of any treatment groups.

4. It was unclear how the authors analyzed data points from the same subjects at different time points and generated the scatter plot in Figure 5. It seems that all the data points were pooled together in this analysis, which would result in an overestimated degree of freedom. A better way to model the data is the linear mixed-effects regression.

Minor points

1. Please add 'eye' as a label to Figure 1. 'N' could be a subject or an eye.

2. 'Vehicle' was not a vehicle or placebo eye drop in this study. Please make it clear.

3. IOP data were not presented, but the authors tested it using an iCare device.

Author Response

Dear reviewer, 

Thank you very much for your time, effort, and comments. It helped us improve the manuscript. In an attempt to make appropriate changes following the your recommendations, we needed to use different software for the statistical analysis which result in the changes of graph types. In other words, the analysis is the same but the graphical representation is different. 

  1. Thank you for your comment. Both groups contain 5 eyes. The fisetin group has only an insignificantly higher age mean than the control (fisetin-free) group and although there are differences in sex distribution, sex doesn’t influence tear production.  As you noted, the small sample size also affected randomization together with all statistics which were applied. 
  2. We agree with your comment. We would like to explain it in more detail. Firstly, there was no previous study with fisetin. For this reason, no sample size calculations were carried out due to the pilot exploratory nature of the study and the unknown vehicle effect, as we state in chapter 4.6. Statistical analysis. Secondly, we wanted to build on the successful in vitro and in vivo studies which are also mentioned in the Bibliography chapter performing the clinical trial. The mentioned studies use a few subjects (3-5) with outstanding results. As we had no benchmark, we decided to choose 5 samples for each experimental group.
  3. Thank you for your point. Healthy control group served as an internal control which helped us understand whether our data are in accordance with literature and whether patients in the experimental groups significantly differ from healthy subjects. We didn’t follow them in time. Therefore, we changed the study design and removed the healthy control group from tear film production and MMP-9 assessment. We adjusted all Figures and text accordingly. Whatsmore, we renamed the “vehicle” group “control” group. 
  4. Thank you for your advice. We conducted a linear mixed effect analysis and adjusted the Figure 5.
  5. We agreed and added “n - number of eyes” to the figure legend.

Reviewer 2 Report

The authors formulated fisetin for topical use.  They evaluated the effect on tear production in dogs with dry eye disease.  They found no difference between the two formulations on tear production.  However, they found some differences on MMP-9.

1. 1     In my opinion, the title is misleading.  I suggest “The effect of fisetin on treatment of dry eye disease in dogs” or something similar.

2.   2   The title say cyclosporine.  However, I see no note of that in the document.  Please advise.

3.     3 A priori, did the authors make any judgement about what is a clinically significant change in any measure.  Further, how did they select sample size?  Typically, that is based upon the clinical significant change and the variability. 

4.      4Related to above, did the authors test a large enough sample to support their conclusion of lack of efficacy.

5.      5There is a larger issue about biomarkers in dry eye.  It seems that MMP9 is not well correlated with dry eye disease – at least tear volume as they measured it in this study. 

Author Response

Dear reviewer, 

Thank you very much for your time, effort, and comments. It helped us improve the manuscript. In an attempt to make appropriate changes following the reviewers’ recommendations, we need to use different software for the statistical analysis which result in the changes of graph types. In other words, the analysis is the same but the graphical representation is different. 

  1. Thank you for your advice. We changed the title to “The Effects of Fisetin on Cyclosporine-Treated Dry Eye Disease in Dogs”.
  2. Thank you for your remark. We reported on a cyclosporine in the 1. Introduction with words: “Therefore, we studied the effect of fisetin addition to cyclosporine-treated canine DED to target oxidative and inflammatory pathways at one time.”, and “This pilot study, for the first time, investigates its complementary therapeutic potential in cyclosporine-treated DED assessing tear production and MMP-9 levels.”, 3.Discussion with words: “This experimental study investigated the effect of fisetin addition to DED treatment with cyclosporine and artificial tears (hypromellose eye drops) on the restoration of tear production and MMP-9 levels in dogs.”, and “This study demonstrates the efficacy of polyphenol fisetin addition to cyclosporine-treated DED.” 4.1 Study design with words: “The randomized, vehicle-controlled, open-label animal trial between fisetin eye drops and the vehicle was performed to evaluate its effect on tear film production and tear film MMP-9 biomarker in dogs treated with cyclosporine.”, 4.2 Participants with the words: “Nine different breed dogs (weight: 8-30 kg) with long-lasting DED (keratoconjunctivitis sicca) treated with cyclosporine eye drops were enrolled in the study.”
  3. Thank you for your question. Clinical significance is certainly a main point in the search for new treatment. However, our study is of a pilot nature. We chose two parameters to monitor - tear production and MMP-9. When speaking of clinical significance, it’s necessary to also monitor tear film stability and corneal surface on a bigger sample for a longer time period. That is beyond the goal of this study, which should examine the fisetin potential.  We intended to have a benchmark as there is no previous study with fisetin and to create more specific hypotheses for future study. 
  4. Thank you for your point. We changed the word “efficacy” in 3.Discussion to “potential”.
  5. Thank you for your comment. MMP-9 is a well established DED biomarker, but the studies are inconsistent about its relation with tear film production (Park JY et al. (2018) Matrix Metalloproteinase 9 Point-of-Care Immunoassay Result Predicts Response to Topical Cyclosporine Treatment in Dry Eye Disease. Transl Vis Sci Technol, 7: 31; Messmer EM et al. (2016) Matrix Metalloproteinase 9 Testing in Dry Eye Disease Using a Commercially Available Point-of-Care Immunoassay. Ophthalmology, 123: 2300-2308; Lanza NL et al. (2016) Dry Eye Profiles in Patients with a Positive Elevated Surface Matrix Metalloproteinase 9 Point-of-Care Test Versus Negative Patients. Ocul Surf, 14: 216-223; Schargus M et al. (2015) Correlation of Tear Film Osmolarity and 2 Different MMP-9 Tests With Common Dry Eye Tests in a Cohort od Non-Dry Eye Patietns. Cornea, 34: 739-744). To better describe the dependence of these two variables, we changed the Figure 5 and the analysis to linear mixed-effects regression.

Round 2

Reviewer 1 Report

The authors have addressed most of my questions.